# Second-Line Chemotherapy in Elderly Patients with Advanced Biliary Tract Cancer: A Multicenter Real-World Study

**DOI:** 10.3390/medicina58111543

**Published:** 2022-10-27

**Authors:** Alessandro Rizzo, Massimiliano Salati, Giorgio Frega, Valeria Merz, Francesco Caputo, Alessandro Di Federico, Andrea Palloni, Riccardo Carloni, Angela Dalia Ricci, Gennaro Gadaleta-Caldarola, Carlo Messina, Andrea Spallanzani, Fabio Gelsomino, Stefania Benatti, Gabriele Luppi, Davide Melisi, Massimo Dominici, Giovanni Brandi

**Affiliations:** 1Struttura Semplice Dipartimentale di Oncologia Medica per la Presa in Carico Globale del Paziente Oncologico “Don Tonino Bello”, I.R.C.C.S. Istituto Tumori “Giovanni Paolo II”, Viale Orazio Flacco 65, 70124 Bari, Italy; 2PhD Program Clinical and Experimental Medicine, University of Modena and Reggio Emilia, 41121 Modena, Italy; 3Division of Oncology, Department of Oncology and Hematology, University Hospital of Modena, 41125 Modena, Italy; 4Osteoncology, Bone and Soft Tissue Sarcomas, and Innovative Therapies, IRCCS Istituto Ortopedico Rizzoli, 40136 Bologna, Italy; 5Department of Medical Oncology, Santa Chiara Hospital, 35127 Trento, Italy; 6Digestive Molecular Clinical Oncology Research Unit, University of Verona, 37129 Verona, Italy; 7Medical Oncology, IRCCS Azienda Ospedaliero-Universitaria di Bologna, Via Albertoni-15, 40138 Bologna, Italy; 8Medical Oncology Unit, National Institute of Gastroenterology, “Saverio de Bellis” Research Hospital, 70013 Castellana Grotte, Italy; 9Medical Oncology Unit, ‘Mons. R. Dimiccoli’ Hospital, Barletta (BT), Azienda Sanitaria Locale Barletta, 76121 Barletta, Italy

**Keywords:** biliary tract cancer, cholangiocarcinoma, second-line, chemotherapy, elderly

## Abstract

Objectives: The ABC-06 and the NIFTY trials recently established the role of second-line chemotherapy (2L) in patients with advanced biliary tract cancer (BTC). Our real-world study aimed to explore 2L in BTC patients aged ≥ 70 years old and to compare their outcomes with younger subjects. Methods: Institutional registries across three academic medical centers were retrospectively reviewed. The Kaplan–Meier methods were used to estimate survival, and the log-rank test was used to make comparisons. Results: A total of 190 BTC patients treated with 2L were identified and included in the analysis. Among them, 52 (27.3%) were aged ≥ 70 years (range 70–87 years). No statistically significant differences in both median overall survival (mOS) and median progression-free survival (mPFS) were recorded between the elderly and younger patients. Absolute lymphocyte count < 1000/mmc (*p* < 0.001) and albumin level < 3 g/dL (*p* < 0.001) were independently associated with worse prognoses. Conclusions: The results of this real-world study suggest that for patients aged ≥ 70 years, 2L could be equally effective for younger patients with survival outcomes aligned to those from the ABC-06 and NIFTY trials. The delivery of 2L should be carefully evaluated and monitored in this patient subset.

## 1. Introduction

Biliary tract cancer (BTC) encompasses a heterogeneous group of rare, aggressive, and poor-prognosis tumors, including gallbladder cancer (GBC), ampulla of Vater carcinoma (AVC), intrahepatic cholangiocarcinoma (iCCA), and extrahepatic cholangiocarcinoma (eCCA), with the latter further subdivided into perihilar (pCCA) and distal cholangiocarcinoma (dCCA) [1,2]. Overall, these tumors account for approximately 10–15% of all primary liver cancers and 3% of all gastrointestinal malignancies [3]. Unfortunately, potentially curative surgical resection is possible only in a small proportion of BTCs at diagnosis since most patients present with locally advanced unresectable or metastatic disease [4,5]. More than ten years after the publication of the landmark ABC-02 trial, the combination of durvalumab plus gemcitabine—cisplatin has recently reported a statistically significant improvement in terms of overall survival (OS) versus placebo plus gemcitabine—cisplatin, with TOPAZ-1 representing the first phase III trial to demonstrate positive results from the addition of an immune checkpoint inhibitor to the reference first-line doublet [6,7]. However, disease progression inevitably occurs during front-line chemotherapy resulting in an overall survival hardly exceeding one year [8]. In the last few years, the phase III ABC-06 trial in non-Asian patients and the phase IIb NIFTY trial in Asian patients demonstrated a statistically significant improvement in progression-free survival (PFS) and OS after cisplatin/gemcitabine failure, establishing modified FOLFOX and liposomal irinotecan plus fluorouracil-leucovorin respectively, as standard-of-care second-line therapies [9,10,11]. More recently, pemigatinib and infigratinib in the molecular subset of FGFR2-rearranged tumors and ivosidenib in IDH1-mutant tumors have become FDA-approved options for previously treated advanced BTC harboring targetable alterations [12].

Geriatric oncology has recently assumed an increasingly important role due to several reasons, including the presence of a growing aging population, especially in Western countries, with some studies suggesting that by 2050 in Europe, there will be around 130 million people aged 65 years old or more (versus 90 million in 2019) [13]. This element will significantly impact medical oncology due to the increase in cancer incidence with age, resulting in a growing number of cancer cases in elderly patients. However, older patients are often under-represented in clinical trials, and study results are seldom reported by age group. Of note, the under-representation of older patients in cancer trials remains an important obstacle to the generation of data on efficacy and safety in this growing patient population [14,15,16]. In several countries, specific geriatric units have been created to help oncologists and geriatricians work together on research, best practice, and continuing medical education [17,18,19]. Although BTC mostly affects the elderly, with a median age at diagnosis of around 66 years, only limited data are available on the optimal management of this population. This real-world study aimed to explore treatment patterns as well as the safety and efficacy of second-line chemotherapy in advanced BTC patients aged ≥ 70 years old, and to compare their outcomes with younger subjects.

## 2. Materials and Methods

### 2.1. Patients

This multicenter, retrospective study was conducted at three Academic Medical Centers in Italy (Bologna, Modena, Verona). Primary eligibility criteria for inclusion were as follows: age of 18 years or above; cyto-histologically proven BTC including iCCA, eCCA, and GBC; unresectable locally advanced or metastatic BTC (according to the 2010 TNM staging system/American Joint Committee on Cancer stage III or IV) [20]; receipt of at least one cycle of second-line chemotherapy. Patients affected by ampullary carcinoma and hepatocellular-cholangiocellular carcinoma were excluded. Patients older than 70 years were considered “elderly” individuals.

Baseline clinicopathological and laboratory data were retrieved from the participating centers’ institutional registries through electronic medical records review. For each BTC patient, the following variables were collected and analyzed from the start of second-line treatment: (1) age; (2) gender; (3) Eastern Cooperative Oncology Group—Performance Status (ECOG-PS); (4) primary tumor site; (5) first-line regimen; (6) second-line regimen; (7) PFS and OS to second-line. In addition, the following hematological and biochemical parameters were collected and analyzed: white blood cell count (cell/µL), neutrophil count (cell/µL), lymphocyte count (cell/µL), eosinophil count (cell/µL), monocyte count (cell/µL), hemoglobin (gr/dl), platelet count (cell/µL), total bilirubin (mg/dL), albumin (g/dL), and carbohydrate antigen 19-9 (CA 19-9) (U/mL); (8) safety data. The study protocol conformed to the ethical guidelines of the 1975 Declaration of Helsinki. The protocol was reviewed and approved by the Area Vasta Emilia Nord Ethics committee (l184/2019).

### 2.2. Statistical Analysis

Median OS was calculated from the date of the first cycle of second-line chemotherapy to the date of death from any cause or last follow-up visit. The individuals that were still alive were censored at the date of their last follow-up or at the cut-off date of 30 November 2020. The secondary endpoint was median PFS, calculated from the date of the first cycle of second-line chemotherapy to the date of disease progression or death from any cause. In descriptive statistics, continuous variables were reported as the median and 25–75 percentile, while categorical variables were reported as absolute and percentage frequencies. Laboratory variables initially recorded as continuous parameters were later dichotomized according to clinical thresholds reported in the literature or according to the median. The objective assessments of tumor response were performed using Response Evaluation Criteria in Solid Tumors (RECIST version 1.1), while adverse events were reported using the NCI Common Terminology Criteria for Adverse Events (CTCAE version 5.0).

The Kaplan–Meier methods were used to estimate survival, and the log-rank test was used to make comparisons [21]. The median follow-up time was calculated using the reverse Kaplan–Meier method. The predictive performance of each covariate on OS was first evaluated by means of the Cox proportional hazard univariate model, selecting those variables with a *p*-value < 0.05 for multivariate analysis. For all tests, a two-sided *p*-value < 0.05 was considered to be statistically significant, with a confidence interval of 95% (95% CI) [22]. The statistical analyses were performed using the SPSS software (version 26; SPSS Inc., Chicago, IL, USA).

## 3. Results

### 3.1. Patients’ Characteristics

A total of 190 advanced BTC patients treated with second-line chemotherapy and fulfilling the abovementioned criteria were identified between October 2002 and November 2020 and were included in the analysis. Among them, 52 (27.3%) patients were aged ≥ 70 years (range 70–87 years), of whom 25 patients had iCCA (48.1%), 15 eCCA (28.8%), and 13 GBC (25%); 56% (*n* = 29) of them were female. Among younger patients (*n* = 138), 71 (51.4%) were female and 85 (61.5%) had ECOG-PS 0-1; iCCA was the most common primary tumor site (*n* = 69, 50.0%), followed by eCCA (*n* = 41, 29.7%). Table 1 summarizes the baseline features of the included patients.

### 3.2. Treatment Patterns

As regards front-line treatment, gemcitabine plus oxaliplatin was the most common first-line regimen in elderly patients (*n* = 22, 42.3%), followed by cisplatin plus gemcitabine (*n* = 14, 26.9%) and gemcitabine monotherapy (*n* = 8, 15.4%). The performance status of elderly patients at the time of first-line chemotherapy start was ECOG-PS 0 in 30 patients (57.7%), ECOG-PS 1 in 17 (32.7%) and ECOG-PS 2 in 5 (9.6%). Among the younger patients, cisplatin plus gemcitabine was the most frequently administered first-line regimen (*n* = 41, 29.7%), followed by gemcitabine plus oxaliplatin (*n* = 39, 28.3%) and gemcitabine monotherapy (*n* = 10, 7.2%). In this age subgroup, the ECOG PS at the time of first-line treatment initiation was 0 in 64 patients (42.1%), 1 in 49 (32.2%), and 2 in 25 (25.7%) patients.

Among elderly patients, the most commonly administered second-line regimens were capecitabine monotherapy (*n* = 14, 26.9%), followed by single-agent gemcitabine (*n* = 8, 15.4%) and gemcitabine plus capecitabine combination (*n* = 8, 15.4%); mFOLFIRI was administered in seven patients (13.5%), mFOLFOX in five (9.6%), cisplatin plus gemcitabine in four (7.7%), XELOX in two (3.8%), docetaxel in two (3.8%), sorafenib in one (1.9%), and XELIRI in one (1.9%). On the other hand, mFOLFIRI was the most frequent second-line treatment in younger patients (*n* = 33, 23.9%), followed by mFOLFOX (*n* = 31, 22.5%) and gemcitabine–capecitabine (*n* = 17, 12.3%) and capecitabine alone (*n* = 16, 11.6%). Ten (7.2%) patients received gemcitabine, while 5-Fluorouracil monotherapy and docetaxel were administered in nine (6.5%) and eight (5.8%) patients, respectively.

### 3.3. Efficacy Outcomes, Safety, and Prognostic Factors

Regarding the antitumor activity of second-line chemotherapy, the disease control rate was 28.8% (*n* = 15) and 29.7% (*n* = 41) in elderly and younger patients, respectively. This was mainly achieved through disease stabilization (*n* = 12, 23.1%, and *n* = 36, 26.1%, respectively), while objective responses were uncommon, with partial responses observed in three (5.7%) and five (3.6%) patients, respectively. Instead, no complete responses were recorded. As concerns the impact of second-line treatment on survival outcomes, no statistically significant differences between the two age groups were seen in both median OS (5.7 months in the elderly versus 6.1 months in younger patients, HR 0.97; *p* = 0.86) (Figure 1) and median PFS (4.7 vs. 4.8 months, HR 0.88; *p* = 0.44) (Figure 2).

Regarding the safety profile of second-line chemotherapy, grade 3 or 4 treatment-related toxicities occurred more frequently in the elderly group (48.5% vs. 8.2%; OR 6.31; *p* < 0.001). In this patients’ subset, myelotoxicity was the most commonly observed grade 3–4 adverse event (29.7%: anemia 15.3%, neutropenia 8.9%; thrombocytopenia 5.5%), followed by fatigue (11.3%) and diarrhea (5.7%). Myelotoxicity was also the most frequently reported grade 3–4 adverse event in younger patients (*n* = 8, 5.8%). Neither cases of permanent treatment discontinuation nor deaths related to second-line chemotherapy were observed in both groups.

When looking at the prognostic significance of major pre-treatment variables in elderly patients, ECOG-PS (*p* = 0.4), hemoglobin (*p* = 0.108), eosinophil count (*p* = 0.811), neutrophil count (*p* = 0.63), monocyte count (*p* = 0.062), platelet count (*p* = 0.695), total bilirubin (*p* = 0.321), and CA 19-9 (*p* = 0.127), were not predictors of OS at the univariate analysis. Notably, absolute lymphocyte count < 1000/mmc (*p* < 0.001) and albumin level < 3 g/dL (*p* < 0.001) were predictor of worse prognosis at the univariate analysis (Figure 3). At the multivariate analysis, lymphocyte count < 1000/mmc (*p* < 0.001) and albumin level <3 g/dL (*p* = 0.03) retained prognostic significance.

## 4. Discussion

When treating elderly cancer patients, medical oncologists are vexed by several challenges [15,16,17]. Firstly, the lack of data from clinical trials represents a long-standing problem in this setting since inclusion and exclusion criteria are usually very strict when it comes to age and the decision-making in the clinic hinges on physician discretion rather than being evidence-based. Consequently, most clinical trials, including younger and fitter cancer patients, do not reflect the patient population we are more likely to face in clinical practice [18,19]. Secondly, the “category” of patients aged ≥ 70 years is widely heterogeneous with various life expectancies, encompassing more fit as well as more frail subjects, with a survival rate that significantly varies from a few months to several years and also depends on the primary cancer type and general health conditions. Based on these premises, we have recently witnessed growing attention paid to using tools to evaluate older patients with cancer. To this end, the implementation of the Comprehensive Geriatric Assessment (CGA) is recommended by several international guidelines, such as ASCO and NCCN [23,24,25]. Unfortunately, these issues are even more important in a rare disease with few treatment options like BTC. If the outcome of BTC patients remains poor, with a five-year overall survival rate of less than 20%, previous studies have reported that elderly patients with advanced BTC may have an even worse prognosis. Several reasons may contribute to this [26,27,28,29,30]. Among these, clinicians are frequently reluctant to choose aggressive systemic treatment for older subjects because of their age, impaired organ function and comorbidities. 

Our study presents its strengths and limitations. Among the former, this represents the first real-world experience generated during routine clinical practice on the pattern of care and treatment outcome of elderly BTC patients undergoing second-line chemotherapy. Furthermore, our study includes a relatively large number of patients enrolled across three highly experienced academic centers, which serve as referral Centers for managing BTC. Besides, a comprehensive collection of patient- and disease-related factors with potential prognostic value has been performed, including clinicopathologic and biochemical parameters. Limitations to be acknowledged are the retrospective nature of the study and the potential selection bias generated by the inclusion of elderly subjects with preserved organ function and good ECOG PS who received treatment in academic centers. Moreover, since these patients were followed before the publication of ABC-06 and NIFTY, treatment sequences were heterogeneous, and some patients received similar first- and second-line treatment (for example, gemcitabine-based treatment as front- and second-line therapy). Conversely, our study did not include BTC patients who followed stop-and-go strategies. Consequently, our findings should be interpreted with caution and require more prospective validation, especially considering the length of the study period.

The results of our real-world study suggest that for patients aged ≥ 70 years, second-line chemotherapy could be equally effective for younger patients with disease control rates and survival outcomes aligned to those from the ABC-06 trial and NIFTY trial [9,11]. As expected, elderly subjects were more likely to experience grade 3–4 treatment-related adverse events (48.5% vs. 8.2%; OR 6.31; *p* < 0.001) in our analysis. Based on the significantly higher incidence of toxicities, the delivery of second-line chemotherapy should be carefully evaluated and monitored in this patient subset. For example, tools like the comprehensive geriatric assessment (CGA)—which represents the gold standard in this setting—as well as the Cancer and Aging Research Group Toxicity Tool (CARG-TT) are fundamental to determining suitability for chemotherapy during the assessment of elderly patients with cancer, including those with BTC [31,32]. Our study presents some similarities and differences with recently published studies regarding elderly patients with BTC [33,34]. A Canadian study conducted by Horgan and colleagues included 321 BTC patients aged ≥ 70 years, older patients were more likely to receive the best supportive care alone than younger subjects [33]. In this report, the authors suggested a similar survival benefit from systemic treatment in older and younger BTC patients. Although the survival advantage of second-line chemotherapy in the elderly remains convincingly established in clinical trials of BTC and despite the limitations affecting our analysis, the study suggests that patients aged ≥ 70 years may derive a similar benefit from second-line treatment than their younger counterparts. Since this comes at an increased risk of toxicities, the decision on whether proceed to second-line chemotherapy should be carefully evaluated in this patient population. Regarding the lack of validated factors, immunological and nutritional parameters, such as lymphocyte count and serum albumin, could aid in selecting elderly BTC patients more fit for further treatment after first-line failure. In another report published by Takahara and colleagues, the authors observed that the efficacy of second-line therapy was modest regarding tumor response. However, it was associated with improved survival [34]. At the same time, Takahara et al. reported that clinical outcomes in first-line settings appeared to affect those of patients treated with second-line chemotherapy [34].

The results of our experience highlight the importance of a proper risk-benefit assessment in this setting. At the same time, it is worth noting that most elderly BTC patients experiencing these events are fragile, pretreated patients, while younger subjects tend to present better bone-marrow function. Moreover, we identified pre-treatment lymphocytopenia and hypoalbuminemia as a potential predictor of poor survival in our analysis, in keeping with the growing body of evidence on the prognostic role of inflammatory, immune, and nutritional parameters in both first- and second-line settings [35]. In addition to well-known clinical-pathologic factors, these could assist in the risk-stratification and treatment decision of elderly BTC in clinical practice. Future trials incorporating multidimensional geriatric assessment tools are warranted in this setting to personalize the care and improve the outcome of this frail subset of patients.

## 5. Conclusions

The results of this real-world study suggest that for patients aged ≥ 70 years, second-line chemotherapy could be equally effective for younger patients, with survival outcomes aligned to those from ABC-06 and NIFTY trials. The delivery of second-line treatment should be carefully evaluated and monitored in this patient subset.

## Figures and Tables

**Figure 1 medicina-58-01543-f001:**
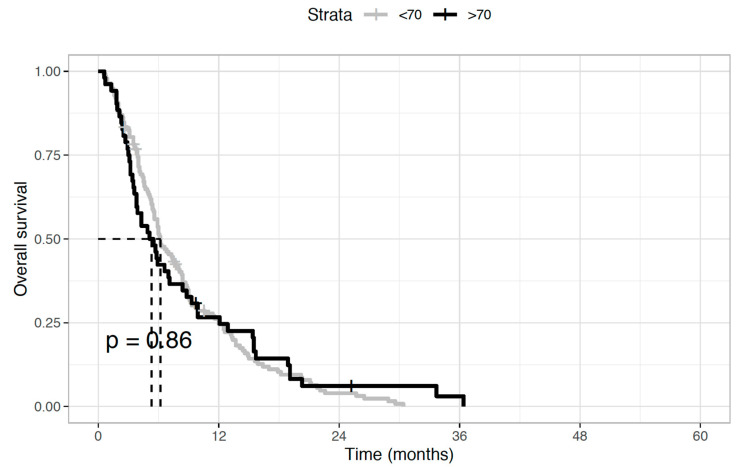
Overall survival (OS) of biliary tract cancer patients receiving second-line chemotherapy. Median OS was 5.7 months in elderly patients and 6.1 months in younger patients (Hazard Ratio 0.97; *p* = 0.86).

**Figure 2 medicina-58-01543-f002:**
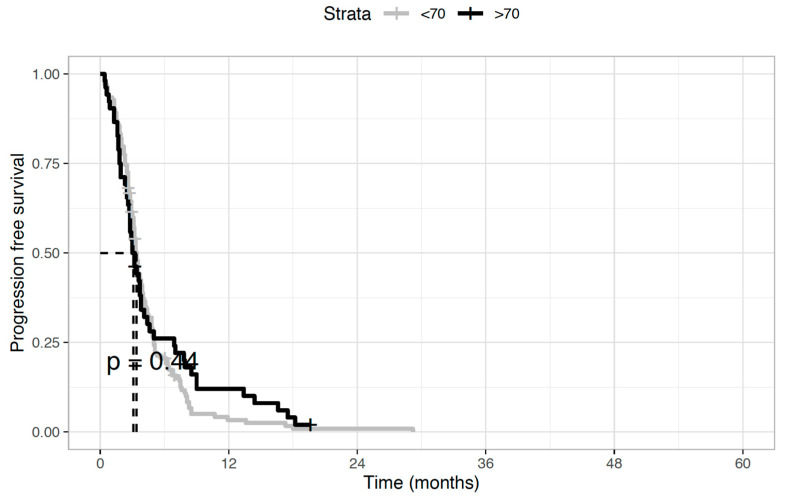
Progression-free survival (PFS) of biliary tract cancer patients receiving second-line chemotherapy. Median PFS was 4.7 months in elderly patients and 4.8 months in younger patients (Hazard Ratio 0.88; *p* = 0.44).

**Figure 3 medicina-58-01543-f003:**
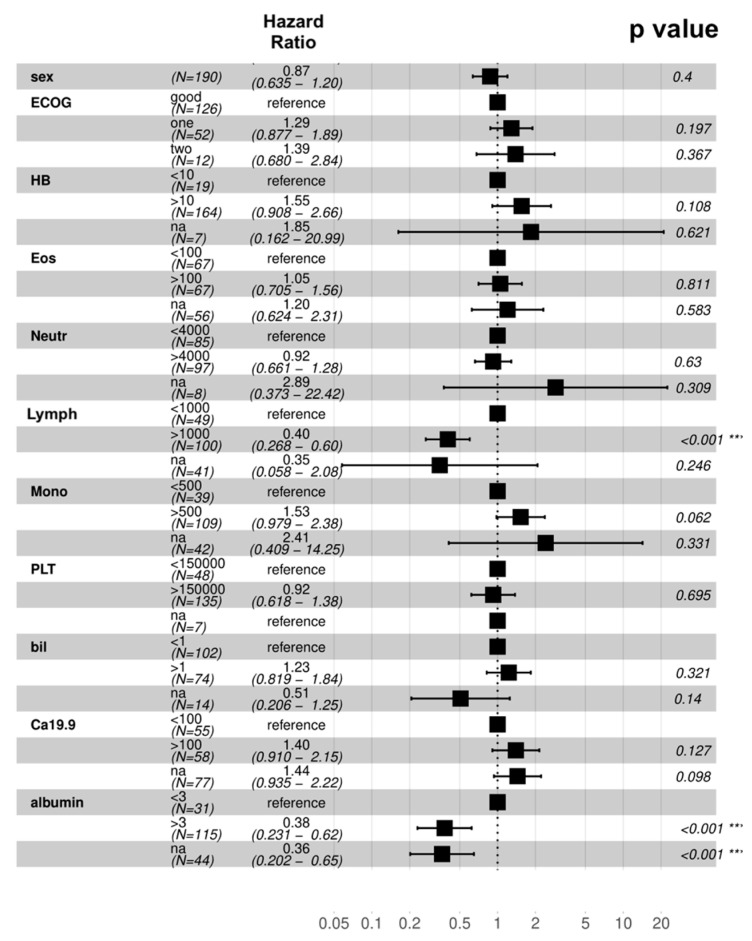
Univariate analysis of overall survival. Abbreviations: Bil—total bilirubin; Eos—eosinophil count; HB—hemoglobin; Lymph—lymphocyte count; Mon—monocyte count; Neutr—neutrophil count; PLT—platelet count. *** statistically significant results.

**Table 1 medicina-58-01543-t001:** Baseline patient characteristics (*n* = 190). Abbreviations: eCCA—extrahepatic cholangiocarcinoma; GBC—gallbladder carcinoma; iCCA—intrahepatic cholangiocarcinoma.

Variable	Patients Aged ≥ 70 Years (*n* = 52)	Patients Aged < 70 Years (*n* = 138)
Age, years median (range)	76 (70–87)	64 (35–69)
Gender Female Male	28 (53.9) 24 (46.1)	71 (51.4) 67 (48.6)
ECOG-PS 0 1 2	10 (19.2) 28 (53.9) 14 (26.9)	14 (10.1) 71 (51.4) 51 (38.5)
Primary tumor site iCCA eCCA GBC	25 (48.1) 15 (28.8) 13 (25.0)	69 (50.0) 41 (29.7) 28 (20.3)
Second-line chemotherapy regimen	Capecitabine: 14 (26.9) Gemcitabine: 8 (15.4) Capecitabine–Gemcitabine: 8 (15.4) mFOLFIRI: 7 (13.5) mFOLFOX: 5 (9.6) Cisplatin–Gemcitabine: 4 (7.6) XELOX: 2 (3.8) Docetaxel: 2 (3.8) Sorafenib: 1 (1.9) XELIRI: 1 (1.9)	mFOLFIRI: 33 (23.9) mFOLFOX: 31 (22.5) Capecitabine–Gemcitabine: 17 (12.3) Capecitabine: 16 (11.6) Gemcitabine: 10 (7.2) 5-Fluorouracil: 9 (6.5) Docetaxel: 8 (5.8)
Third-line chemotherapy	10 (19.2)	33 (23.9)

## Data Availability

Not applicable.

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
