# Peer review of "Second-Line Chemotherapy in Elderly Patients with Advanced Biliary Tract Cancer: A Multicenter Real-World Study"

_medicina, 2022, doi:10.3390/medicina58111543_

Round 1

Reviewer 1 Report

Overall I think this is a simple yet important study delineating the outcomes of elderly patients with poor prognosis cancer. Research in this area is vital to improve awareness and knowledge in the need for a more tailored approach to our elderly patients and I am encouraged to see a study that addresses the use of second line treatment. I have a few suggestions for improvements as below. Well done to the authors in putting this together.

Part 1.

·       Lines 68-70.

Would be useful to reference this statement re underrepresentation and infer why it is important they are included i.e., heterogenous population and need for clinical data in frailer group, changes in host tumour interaction and tumour biology with age etc. Also, to discuss international guidelines e.g., SIOG geriatric oncology priority initiative recognises improving relevance of cancer trials for an elderly population as one of their key priorities

Part 3.1

·       Line 122

‘Among them, 27.3% (n=52) patients’ should be 52 patients (27.3%).

Part 3.2

·       Lines 132- 142

I am not sure that this point is relevant about PS at first line therapy unless the authors can demonstrate that elderly patients are statistically more likely to have a change in PS following first line treatment as compared to younger. Not sure if helpful to include otherwise.

Part 3.3

·       Was there a statistically significant difference in disease control rate between cohorts?

·       It is not noted in the variables collected in the method section that you also collected toxicity data, should include this.

·       It would also be interesting to look at the difference in outcomes and toxicity between elderly patients dosed with doublet regimen in second line setting compared to those dosed  with single agent regimen. As you already have the data collected this should not be too burdensome. Of course, selection bias will be at play with more frail patients being selected for single agent therapy, but possibly less so than the selection bias of examining patients who were selected for treatment as compared to those who were not.

·       Line 156-157

Presume this refers to elderly and younger patients respectively, reclarify here.

Part 4.

·       Lines 219-233

Some references of inclusion of elderly patients in clinical trials would be useful.

·       Have you looked at inclusion of elderly patients in second line studies for BTC i.e., ABC 06/NIFTY and second line biomarker linked studies? It would be good to comment on inclusion of elderly patients in these studies (I think most do not have an age cut off) and see if they report on outcomes in elderly as this will sometimes be available in supplementary information. It would be nice to put some current context on this. Even though elderly patients may be more likely to be included at this point, the proportion of elderly patients included are often not reflective of real world incidence and there is a lack of integration of geriatric assessment tools and virtually no elderly only clinical trials.

·       Lines 257-270

Aside from assessing laboratory parameters as suggested, can you suggest other ways that clinicians can determine risk of toxicity from treatment in elderly patients? Obviously CGA is the gold standard, but many institutions do not yet have a fully supported geriatric oncology service. I think given the findings of your study, which show quite a striking difference in severe toxicity between groups, it would be prudent to mention tools like the CARG toxicity tool. This would be useful as a simple measure that could be integrated into the assessment of an elderly patient when determining suitability for chemotherapy (does not replace CGA ).

·       Lines 285-289

This study by Takahara et al needs to be referenced. Interestingly this study shows that effect of second line therapy was more prominent in elderly which you should mention. This may be better placed earlier in the body of discussion when discussing clinical efficacy of treatment in elderly patients compared to younger.

Author Response

Dear Reviewer, thank you for your comments and for the time spent revising our paper.

  1. Thank you for this comment. We modified some parts of the introduction section, expanding some paragraphs regarding current issues about the inclusion of elderly patients in cancer clinical trials. In addition, we added some references in the specific section, including the one regarding specific international guidelines regarding this topic as well as key papers. All our changes have been highlighted in purple color.
  2. Thank you for catching this oversight, that we have now corrected in the text (abstract and results section). Our changes have been reported in orange.
  3. Thank you for this comment. We removed this information in the revised paper.
  4. No significant difference in terms of disease control rate was observed. In particular, the disease control rate was 28.8% (n=15) and 29.7% (n=41) in elderly and youngers, respectively, as reported in section 3.3.
  5. Thank you, we modified accordingly.
  6. Thank you for this comment. As regards your question, as reported in Table 1, younger patients were mainly treated with combination chemotherapies as second-line treatment (mFOLFIRI, mFOLFOX, and gemcitabine plus capecitabine – 23.9%, 22.5%, and 12.3%, respectively), while the most commonly second-line therapies for elderly BTCs were capecitabine monotherapy (26.9%), gemcitabine (15.4%), and gemcitabine plus capecitabine (15.4%). Thus, elderly patients were mainly treated with single-agent chemotherapy, and we did not perform this analysis because of two reasons: the presence of important selection bias, as you correctly state, and the overall small sample size. Thank you again for your comment.
  7. We better specified this point, as suggested (grey).
  8. We modified accordingly.
  9. Thank you for this question. We searched in the full text and the supplementary appendix of ABC-06 and NIFTY trials, and we did not find specific analyses regarding elderly patients. At the same time, the two studies included a heterogeneous patient population in terms of age; for example, median age in ABC-06 was 65 years, with a range from 59 to 72 in the experimental and the control group.
  10. We better specified this point, as suggested. In particular, we added the following section (red):

“Based on the significantly higher incidence of toxicities, the delivery of second-line chemotherapy should be carefully evaluated and monitored in this patient subset. For example, tools like the comprehensive geriatric assessment (CGA) – which represents the gold standard in this setting – as well as the Cancer and Aging Research Group Toxicity Tool (CARG-TT) are fundamental to determine suitability for chemotherapy during the assessment of elderly patients with cancer, including those with BTC.”

  1. Thank you for this suggestion. We moved the study published by Takahara earlier in the body of the Discussion section and we modified some sentences (green).

Thank you again for your comments. We hope the revised paper will better suit the journal.

Reviewer 2 Report

The authors detail a multicenter retrospective study in second line chemotherapy for advanced biliary tract cancer. This is an important topic with scarce evidence. However, the heterogeneity of the treatment given makes it difficult to reach any solid conclusions. My concerns are detailed below.

Major points:

1.       As the authors state in their limitations, the sample is very heterogeneous. It is difficult to conclude from this study that “second- line chemotherapy could be equally effective as for youngers” (lines 293-294) because the groups received different types of treatment. Were older patients more likely to receive monotherapy, lower relative dose intensities, etc?

2.       Was PFS considered? If so, please include relative analyses. Please also indicate how many went on to receive third-line thearpy, if any.

3.       How was the 2L chemotherapy regimen selected? Physician’s discretion?

 Minor points:

1.       Lines 43-46: Although I understand the authors excluded ampullary cancer from the study, I believe ampullary cancer is also included in the definition of BTC.

2.       Lines 110-117: Did the authors use the log-rank test? If so, please state.

3.       3.2 Treatment patterns: perhaps first-line treatment should be discussed before second-line therapy.

4.       Figure 3: The authors label the figure “subgroup analysis” but it looks to me like univariate Cox analysis. The box and whisker plots are probably not necessary, particularly for the reference value.

Author Response

Dear Reviewer, thank you for your comments and for the time spent revising our paper.

Major comments

  1. Thank you for this comment. We modified the sentence above (green), reporting that our results are burdened by the limitations of the current study. As regards your question, as reported in Table 1, younger patients were mainly treated with combination chemotherapies as second-line treatment (mFOLFIRI, mFOLFOX, and gemcitabine plus capecitabine – 23.9%, 22.5%, and 12.3%, respectively), while the most commonly second-line therapies for elderly BTCs were capecitabine monotherapy (26.9%), gemcitabine (15.4%), and gemcitabine plus capecitabine (15.4%).
  2. We included the PFS analysis (Figure 2) and we included this information in the revised table. In particular, 19.2% and 23.9% of elderly and young patients received third-line chemotherapy.
  3. Thank you for your question. Yes, since the study was a multicenter, retrospective report conducted at three different medical Italian centers, the systemic third-line chemotherapy was chosen at physician’s discretion.

Minor comments

  1. Thank you for this suggestion. We modified accordingly, in the Introduction section.
  2. We better specified this point (green). Thank you for catching this oversight.
  3. Thank you for this suggestion. We modified accordingly in the revised paper.
  4. We modified accordingly.

Round 2

Reviewer 2 Report

The authors have adequately revised their manuscript. I have no further comments.